# Implications for food safety of the size and location of fragments of lead shotgun pellets embedded in hunted carcasses of small game animals intended for human consumption

Rhys Green[1,2]*, Mark Taggart[3], Deborah Pain[2,4], Keturah Smithson[5]

1 RSPB Centre for Conservation Science, Bedfordshire, United Kingdom, 2 Conservation Science Group, Department of Zoology, University of Cambridge, Cambridge, United Kingdom, 3 Environmental Research Institute, University of the Highlands and Islands, Thurso, United Kingdom, 4 School of Biological Sciences, University of East Anglia, Norwich, United Kingdom, 5 Cambridge Biotomography Centre, Museum of Zoology, University of Cambridge, Cambridge, United Kingdom

* reg29@cam.ac.uk

**Data Availability Statement:** All relevant data are within the paper.

## Abstract

Carcasses of common pheasants (*Phasianus colchicus*) killed by hunters using shotguns are widely used or sold in the United Kingdom and elsewhere for human consumption. Almost all of the birds are shot using shotgun pellets composed principally of lead (Pb). Lead shotgun pellets often fragment on impact within the bodies of gamebirds, leaving small lead particles in the meat that are difficult for consumers to detect and remove and from which a greater proportion of lead is likely to be absorbed. Chronic exposure to even low levels of lead is associated with negative health effects in humans and especially in groups particularly vulnerable to the effects of lead, which include young children and pregnant women. Our study used a high-resolution computerised tomography X-ray scanner to locate, in three dimensions, metal fragments embedded within carcasses of eight pheasants sold for human consumption in the UK. Small radio-dense fragments (<2 mm diameter), assumed to be metallic lead, were present in all of the pheasant carcasses examined (mean number: 39 per carcass) and many were too small (<0.1 mm diameter) and too distant from the nearest large shotgun pellet for it to be practical for consumers to detect and remove them without discarding a large proportion of otherwise usable meat. Consumers of carcasses of pheasants killed using lead shotgun ammunition are likely to be exposed to elevated levels of dietary lead, even if careful food preparation is practiced to remove shotgun pellets and the most damaged tissue.

## 1. Introduction

The meat of wild-shot game animals killed using lead ammunition is widely eaten by humans and many individuals eat it frequently. In the European Union (EU) and the United Kingdom (UK) combined, it has been estimated that about five million people eat at least one meal of game meat per week, averaged over the year [1]. Each year, consumers in the UK eat about

**Funding:** The only funding obtained for this study was to pay for time on the University Museum of Zoology's micro-CT facility. The UK environmental charity the Royal Society for the Protection of Birds (RSPB) provided funds for this purpose. There was no grant number. The funder had no role in study design, data collection and analysis, decision to publish, or preparation of the manuscript.

**Competing interests:** The authors have declared that no competing interests exist.

11,000 tonnes of meat derived from wild-shot gamebirds [2]. Much of this is from carcasses of the common pheasant (*Phasianus colchicus*), which is the bird species most commonly shot for human consumption in the UK [3]. Virtually all pheasants shot in the UK and available for human consumption there are killed using shotgun ammunition made from lead, rather than from other metals [4,5], and this is also the most frequently used type of shotgun ammunition globally. Lead is a toxic non-essential element in vertebrate animals. Absorbed lead has adverse effects on human health with the potential to impact most body systems. Some effects on humans occur in individuals with chronic low-level exposure and low concentrations of lead in blood [6]. Previously, it appears that it was widely assumed that almost all metallic lead embedded in the carcass of a game animal shot using lead ammunition was in the form of intact or almost intact projectiles and that detection and removal of these from food during preparation or by the consumer prior to swallowing would result in the quantity of ammunition-derived lead absorbed being negligible [7]. As far as we are aware, these assumptions were not made explicit in decision-making about public health, but we note the absence of ingestion of meat from game shot with lead bullets and lead shotgun pellets as a potential route of exposure to lead in the Codex Alimentarius Code of Practice on reducing exposure to lead from food [8]. This assumed lack of hazard to human health from embedded ammunition-derived lead may also account for the absence of a Maximum Level (ML) for lead in human foodstuffs derived from wild-shot game animals set in the Codex Alimentarius General Standard for Contaminants and Toxins [9] and the European Union's Regulation (EC) No 1881/2006, which sets maximum levels for certain contaminants (including lead) in foodstuffs [7,10]. No MLs were set for lead in game meat under these systems, even though they both set MLs for lead for many other foodstuffs, including meat derived from livestock (cattle, sheep, pigs and poultry) and shellfish harvested from the wild. However, observational studies of humans found a positive correlation between the concentration of lead in blood plasma and the frequency of consumption of the meat of wild animals killed using lead ammunition [11]. This suggests that some lead derived from ammunition is both ingested and absorbed by human consumers. Absorption of lead from ingested particles of the metal has also been demonstrated experimentally in non-human mammals [12,13]. If the apparent assessment of a negligible potential hazard from ammunition-derived lead in game meat by the Codex Alimentarius and the European Union's Regulation (EC) No 1881/2006 is incorrect, this may have negative consequences for public health. Removing small and widely-distributed fragments of ammunition-derived metallic lead from meat is difficult [14], so dietary exposure to ammunition-derived lead and its absorption and significance for public health may be greater than was previously assumed.

Several two-dimensional X-radiography studies have shown that mammals killed using lead rifle bullets often contain lead fragments which are small, numerous and widely dispersed in edible soft tissues at some distance from wound channels. Such studies of red deer (*Cervus elaphus*) [15], roe deer (*Capreolus capreolus*) [15] and white-tailed deer (*Odocoileus virginianus*) [12,16] killed using lead bullets revealed the presence of many small bullet fragments in tissue at distances up to 45 cm from the wound channel. Estimation of fragment size from X-radiographs showed that small fragments comprised a substantial proportion of the total detected fragment mass. Knott et al. [15] found that 34% of the mass of lead embedded in eviscerated deer carcasses was composed of fragments of 10 mg mass or smaller. Standard butchery practices used on hunted deer in the USA were shown not to remove all fragments as small as this. Hunt et al. [12] found at least one bullet fragment in 32% of 234 0.91 kg packages of minced meat prepared by butchers from carcasses of wild-shot deer. Such small fragments in processed meat would probably not be detected by consumers.

Recent two-dimensional X-radiography has similarly shown that lead shotgun pellets also fragment when they are fired into gamebirds and waterfowl, leaving small radio-dense metal fragments, identified as shards of metallic lead. Pain et al. [17] found these small fragments in 76% of X-radiographs of 121 wild-shot gamebirds of six species obtained from selected supermarkets, game dealers or game shoots in the UK. Small metallic fragments have also been observed using two-dimensional X-radiography in carcasses of wild-shot common starlings (*Sturnus vulgaris*) [18] and Eurasian woodcocks (*Scolopax rusticola*) [19]. However, none of these studies quantified the amount, location and size distribution within gamebird carcasses of metallic fragments derived from lead shotgun ammunition. In this paper, we report results from the use of a high-resolution computerised tomography X-ray scanner to locate in three dimensions and to measure the size of metal fragments embedded within the carcasses of wild-shot common pheasants killed using lead shotgun ammunition and sold for human consumption in the UK.

## 2. Material and methods

### 2.1. Compliance with ethics requirements

This study did not involve any animal experiments. Carcasses of dead birds used in the study were purchased from food retailers and were farmed domestic fowl or pheasants legally killed by hunters in the UK.

### 2.2. Preparation of samples and imaging

**2.2.1 Wild-shot pheasants.** We purchased carcasses of eight free-ranging common pheasants, which had been shot by hunters on farmland near Peterborough, Cambridgeshire, UK, in late January 2016, and were on sale to the public at a butcher's shop in Cambridge UK on 6 February 2016. When purchased, the carcasses were fresh, plucked, eviscerated and shrink-wrapped with the heads, wings, tarsi and feet removed, but with the skin left on. We imaged the pheasant carcasses using a Nikon Metrology XT H 225 ST High Resolution CT Scanner, at the Cambridge Biotomography Centre in the University Museum of Zoology, Cambridge UK. With the settings we used, the size of voxels imaged by this scanner was a cube with approximately 0.1 mm sides (i.e., one-thousandth of a cubic millimetre in volume). Fragments of metal within the tissues showed up in scanner images as bright specks, with the smallest ones visible being one voxel in size. A fragment of this size is about ten thousand times smaller than a #6 shotgun pellet, which is about 9 cubic millimetres in volume. The CT scanner with these settings visualises a pheasant carcass as about 1,900 virtual slices, each of which is about 0.1 mm thick. CT images located not only whole or nearly-whole lead shotgun pellets, but also tiny shards of metal.

**2.2.2 Calibration using domestic fowl.** We also purchased two fresh, skin-on, eviscerated carcasses of poussins, which are juvenile domestic fowl (*Gallus domesticus*), from a Waitrose and Partners supermarket in Cambridge UK. We mixed known-size lead spheres with cellulose wallpaper paste and injected this mixture into the pectoral and leg muscles of the domestic fowl carcasses using a hypodermic syringe. We also injected small particles (0.2–1.0 mm diameter) of ground-up domestic fowl leg bone (tibiotarsus) in the same way. We began with the needle of the syringe fully inserted into the muscle and then slowly withdrew the needle whilst gently pressing the syringe plunger. This procedure left a linear trail of lead spheres within the muscle. For each injection, we used a batch of lead spheres sorted by diameter under a binocular microscope to be of similar size within the batch, but of different sizes across ten batches. Before injection, we measured the total mass of spheres in each batch to ±0.01 mg with an electronic balance and calculated the mean mass $M$ per sphere. The mean mass in mg per sphere

for the ten batches was: 0.12, 0.13, 0.14, 0.16, 0.32, 0.48, 0.56, 0.68, 3.67 and 4.47. We converted these mean mass values in mg to volumes $V$ by dividing $M$ in mg by the specific gravity of lead (11.34) to give $V = M/11.34$. We then estimated the mean sphere diameter $D$ (in mm) from the volume using $D = 2(3V/4\pi)$. We checked that this method was accurate in determining from their measured masses the diameter of five #6 lead shotgun pellets, of which we measured the diameter directly to ±0.01 mm using dial calipers. We recorded which muscle we injected with each batch of spheres or with bone fragments. We then imaged the domestic fowl carcasses in the CT scanner using the same settings as used for the pheasants and obtained the batch mean diameters derived from images of between 4 and 34 spheres from each batch.

**2.2.3 Estimation of fragment size.** We used ImageJ software [20] to visualise the virtual slices through the pheasant and domestic fowl carcasses. Shotgun pellets, metal fragments and spheres embedded in the tissue were clearly indicated by voxels with a much higher intensity than the surrounding tissue. We counted these high-intensity voxels in each slice and summed the numbers of high-intensity voxels associated with the same object in successive slices to obtain a measure of the apparent volume $V$, in cubic millimetres, of each embedded metal object. These volume estimates were also used to estimate the average apparent mean diameter $AD$, in millimetres, of each object as $AD = 2(3V/4\pi)$ under the assumption that the objects were approximately spherical. This procedure would introduce some inaccuracy into the estimation of the mean diameter of non-spherical fragments, but the dimensions of clumps of high-intensity voxels did not suggest any marked asymmetry, so we think that the effect of this is likely to be minor.

## 2.3. Recovery of pellets

After all scanning was complete, we cut up each pheasant carcass and placed all pieces from each bird individually into a bag made of terylene fabric and suspended all bags in a water bath with a stirrer containing biological enzyme washing powder (Biotex) dissolved in water at 40˚C. The washing powder solution was replaced every few days. We recovered all shotgun pellets seen on the CT scans and some of the larger metal fragments from the individual bags after 12 days of digestion. We obtained metal pieces from seven of the eight pheasants and analysed 29 of those pieces to identify the metal of which they were principally composed. The pieces were weighed to ±0.01 mg and tested with a bar magnet to see whether they were attracted to it. Each piece was placed into a separate 50 ml polypropylene tube, to which 5 ml of trace metal grade concentrated nitric acid was added. The material was left to dissolve for about seven days at room temperature. After this time, the tubes all contained a white precipitate and no visible metal particles. The volume of liquid in each tube was made up to the 50 ml mark with Milli-Q Type I water, whereupon the precipitate entirely dissolved. Solutions were then serially diluted and analysed using an Inductively Coupled Plasma Atomic Emission Spectrometer (ICP-AES; Varian 720ES with SPS3 autosampler) to identify the principal metal elements present. The composition by mass of these elements in each sample was estimated and expressed as a proportion (%) of the initial mass of the metal fragment. Further details of the ICP-AES procedure are given elsewhere [4].

## 3. Results

### 3.1. Composition of metal fragments recovered from pheasant carcasses

The 29 metal fragments and pellets recovered from seven of the eight pheasant carcasses ranged in mass from 6 mg to 155 mg. No fragments were recovered from one carcass with no detectable whole shotgun pellets visible on the CT scans, though scans of this carcass did reveal visible small metal fragments. None of the recovered metal pieces was attracted to a magnet, which indicates that they were not derived from steel shotgun pellets. The mean of the

estimated proportion of mass determined as consisting of lead by ICP-AES analysis across all metal pieces was 99.9%, with the lowest value being 69.0%. We conclude that all of the recovered shot and fragments were composed principally of metallic lead.

## 3.2. Appearance in the CT scans of bone fragments injected into domestic fowl carcasses

The bone fragments injected into domestic fowl muscle were visible on the CT scans but were clearly distinguishable from metal fragments. Even the smallest metal fragments appeared as bright white spots on the CT images and this contrasted strongly with the image of the surrounding soft tissue. Small bone fragments showed much less contrast and had a greyish appearance. The same differences in appearance were evident on the CT scans of pheasant carcasses (Fig 2).

## 3.3. Relationship between the apparent diameter of lead spheres injected into domestic fowl carcasses, as determined by the CT scanner, and their actual diameter and mass

Variation among batches in the mean apparent diameter *AD* of lead spheres injected into domestic fowl carcasses, as determined by the CT scanner, was closely correlated with the diameter *D* calculated from the measured mass and volume of the spheres ($r$ = 0.972; Fig 1). We expected *a priori* that *AD* and *D* would be directly proportion to each other and therefore that a least-squares regression constrained to pass through the origin ($D = cAD$) would give a good description of the relationship for prediction of *D* from observed values of *AD* determined from images of fragments in pheasant carcasses. This expectation was supported by a comparison of the residual sums of squares from this model with that from the ordinary least-squares regression model in which both the intercept *a* and the slope *b* were estimated ($D = a + bAD$), with $a$ = -0.0981 and $b$ = 0.854. Comparison of the two models indicated that inclusion of the intercept term *a* did not result in a significant improvement of model fit ($F_{1,8}$ = 3.66, $P$ = 0.092). The estimate of *c* for the regression model constrained to pass through the origin was 0.724 (standard error = 0.031).

The CT-derived mean diameter (*AD*) was always considerably larger than the mean diameter derived from measured mass (*D*), with the regression model described above indicating that *AD* was, on average a factor of $1/c$ (= 1.38) larger than *D*. We attribute this discrepancy to a blooming artefact; a phenomenon frequently observed in micro-tomography, whereby images of radio-dense particles appear to have a halo of bright voxels around them which make them look larger than they really are [21]. The blooming artefact can be seen especially clearly in the CT image of a whole shotgun pellet in a pheasant carcass shown in Fig 2.

We also modelled the mass of metal particles in the domestic fowl using an ordinary least squares regression model fitted to data for lead spheres injected into the carcasses. We took natural logarithms of mean measured mass *M* in mg and apparent CT-derived mean diameter *AD* in mm for the ten batches of spheres and fitted the regression with log mass as the dependent variable. The fitted model was $\log_e(M) = 0.823 + 3.352 \log_e(AD)$ (Fig 3). This function was then used to estimate the mass of individual metal fragments and shot in pheasants from their apparent diameters in the CT scans.

## 3.4. Numbers and distribution of shotgun pellets and metal fragments in pheasant carcasses

It is likely that all of the large, apparently near-spherical, radio-dense >2 mm diameter objects we observed on the CT images were whole or nearly-whole shotgun pellets, which we identified by ICP-AES analysis as being composed principally of lead (see section 3.1). The most

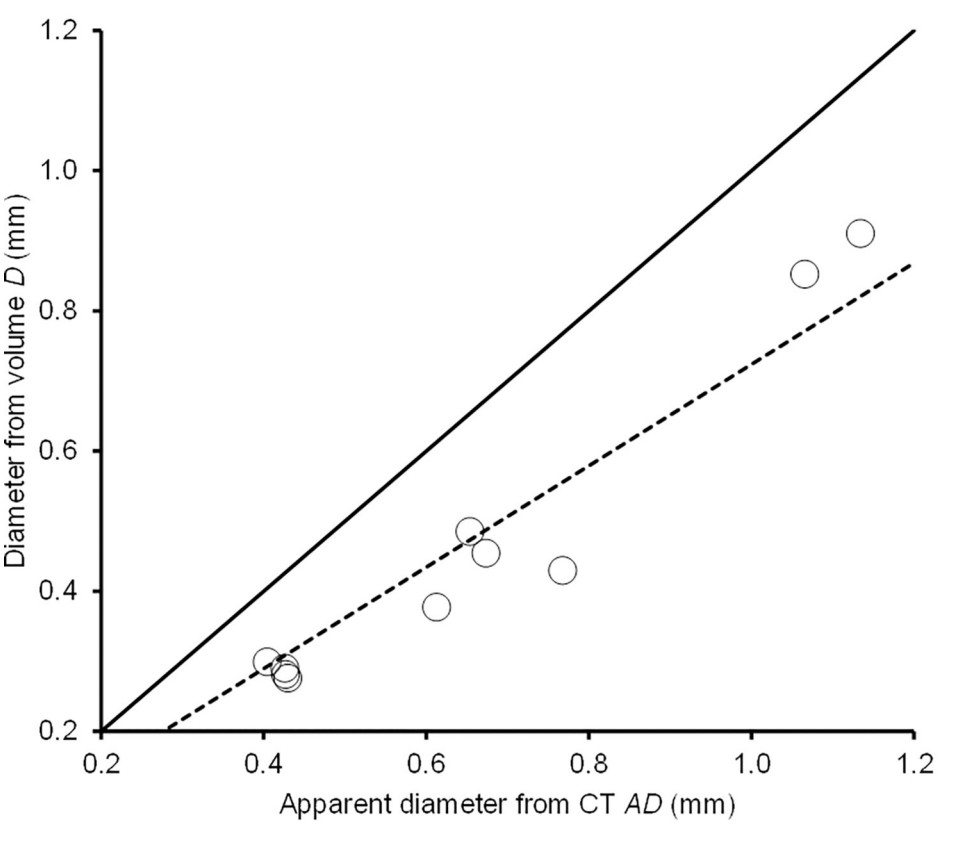

**Fig 1. Known diameter of lead spheres in relation to their apparent size on a micro-CT scan.** Mean diameter $D$ (mm) of ten batches of lead spheres injected into domestic fowl carcasses, derived from their measured mean mass and estimated volume, in relation to the apparent mean diameter $AD$ of the same batches of injected spheres derived from their volume and estimated from the number of radio-dense CT voxels per sphere. The solid line shows the relationship expected if the two diameter estimates were equal. The dashed line shows the regression of $D$ on $AD$ constrained through the origin, $D = 0.724AD$.

frequently used sizes of shotgun pellets used to kill pheasants in the UK are #4, #5, and #6 (diameters 2.6–3.1 mm). The CT images identified a mean of 3.5 shotgun pellets per pheasant carcass (SE = 0.8; range 0 to 7) and a mean of 39.0 smaller (<2 mm diameter) metal fragments per carcass (SE = 10.7; range 3 to 68). There was a non-significant tendency for carcasses with a large number of shotgun pellets also to have a large number of small fragments (Spearman rank correlation coefficient $r_S$ = 0.627; $P$ = 0.114). Three-dimensional plots derived from CT-derived co-ordinates of the metal pieces showed that they were widely distributed within the birds' tissues and that some of the small fragments were far from the nearest pellet (Fig 4). The mean of carcass means of the Euclidean distance between all of the metal fragments in the carcass and the nearest shotgun pellet to each fragment was 24 mm (SE of mean 9 mm; range of carcass means 10–38 mm). The most distant metal fragment from the nearest shotgun pellet, averaged 52 mm away from it (range across carcasses 17–68 mm; SE of mean 20 mm).

### 3.5. Frequency distribution of diameters of metal pieces embedded in pheasant carcasses

We imaged 340 metal objects embedded in the pheasant carcasses, including shotgun pellets with diameters >2mm. Of these, 312 (92%) had estimated diameters $D$ of less than 1 mm. No

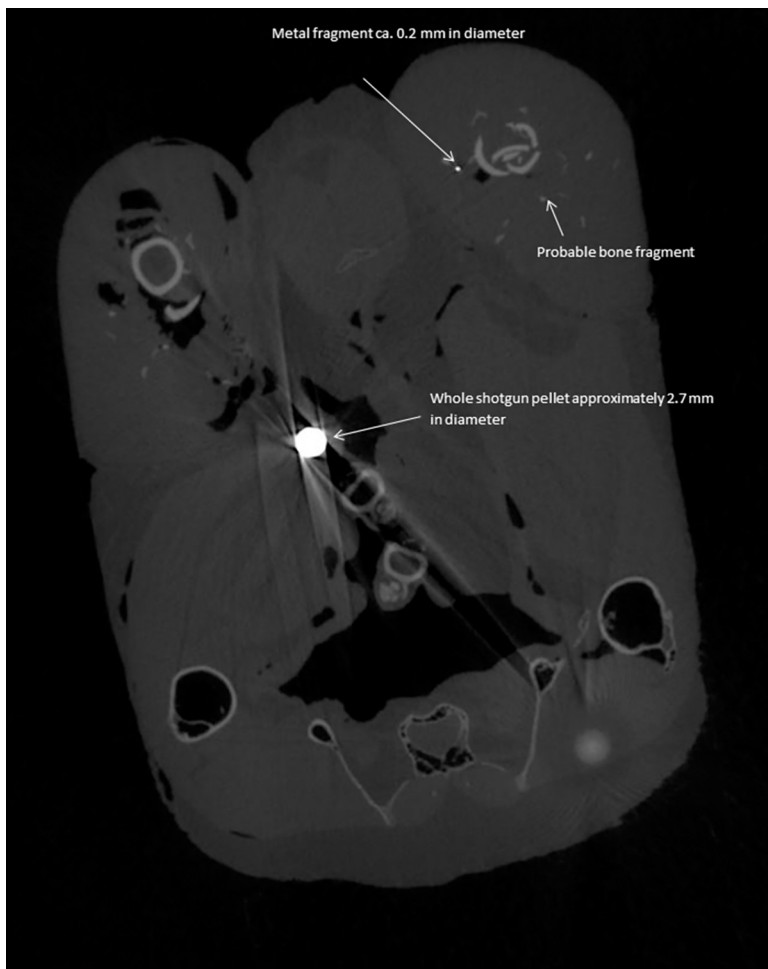

**Fig 2. Micro-CT slice through a pheasant carcass.** This shows a whole shotgun pellet, a small metal fragment and a probable bone fragment. Note the obvious blooming artefact effect around the shotgun pellet.

fragments were detected with diameters in the range 1–2 mm. The probability distribution of the diameters of the 312 small fragments indicated that most were at the smaller end of the observed range, with 86% of the small fragments (268/312) having diameters <0.3 mm (Fig 5). The smallest fragment diameter we measured was 0.07 mm.

### 3.6. Estimated mass of metal pieces embedded in pheasant carcasses in relation to fragment diameter

We assumed that the metal pieces detected in the pheasant carcasses were all composed principally of lead. On this basis, we calculated the cumulative distribution of the estimated mass of lead in each carcass in relation to fragment diameter, after allowing for the effect of the blooming artefact on apparent diameter by multiplying the apparent diameter $AD$ by the regression parameter $c$, estimated as described above. We plotted diagrams showing the cumulative mass of fragments in relation to estimated fragment diameter $D$ for each pheasant carcass. These indicate that a substantial total mass of lead in the form of small fragments was present in the pheasant carcasses (Fig 6). Assuming that all of the fragments smaller than 1 mm diameter were composed of lead and would probably be swallowed by consumers, because they were

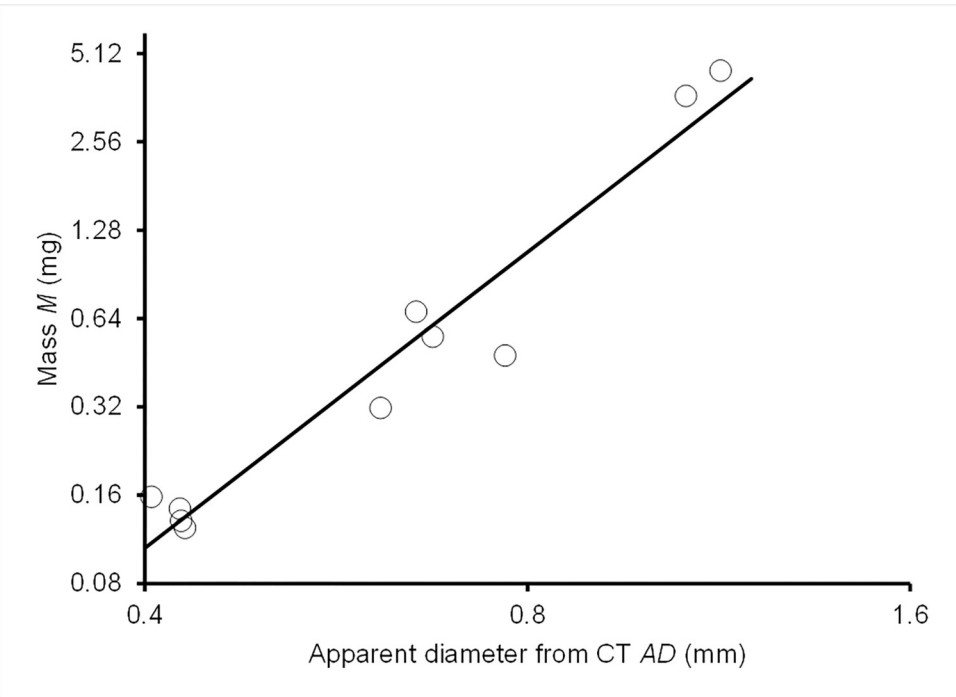

**Fig 3. Known mass of lead spheres in relation to their apparent size on a micro-CT scan.** Measured mean mass $M$ (mg) of ten batches of lead spheres injected into domestic fowl carcasses in relation to the mean apparent diameter $AD$ (mm) of the same batches of spheres derived from CT images. Both axes are logarithmic but the values shown are not transformed. The line is the ordinary least squares regression fitted to the log-transformed values was $\log_e(M) = 0.823 + 3.352 \log_e(AD)$.

too small to be detected and removed, an arithmetic mean of 3.4 mg of lead (SE 1.1 mg) per pheasant consumed would be ingested. There was a non-significant tendency for carcasses with a large number of shotgun pellets also to have a high total estimated mass of lead from fragments smaller than 1 mm (Spearman rank correlation coefficient $r_S = 0.627$; $P = 0.114$).

## 4. Discussion

All of the pheasant carcasses in our study contained embedded small metal fragments, in addition to shotgun pellets, and most contained a considerable number of them. This finding is similar to that of Pain *et al.* [17] who found that 82% of 22 pheasant carcasses sampled had small metal fragments visible on conventional two-dimensional X-radiographs and 14% of birds had more than 15 small fragments. In our study, a much higher proportion of pheasant carcasses (75%) had more than 15 small embedded fragments than in the study of Pain et al. [17]. This is probably because the resolution of the micro-CT scanner was greater than that of the two-dimensional X-ray machine. The minimum estimated fragment diameter we recorded was 0.07 mm, which was the limit of resolution of the CT scanner. Given that many fragments recorded were close to this limit of resolution (86% of small fragments were <0.3mm) it seems reasonable to suppose that smaller fragments than this were also present which we did not detect. It seems likely that all of the fragments we detected were composed of lead, since all of the pieces of metal we recovered from the carcasses were found to be composed of lead. In addition, the non-significant tendency for carcasses with the highest number of shotgun pellets, which were demonstrated to be composed of lead, to also have the highest number of small fragments detected on the CT scans suggests that the small fragments were derived from

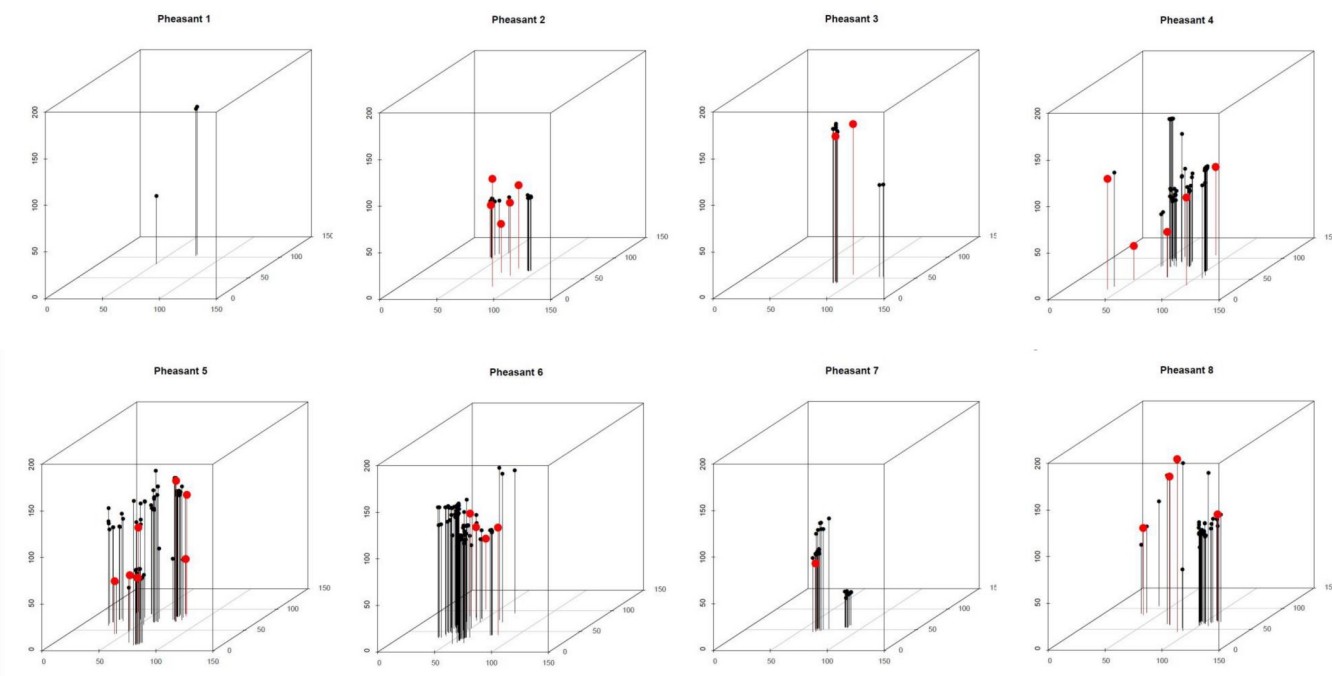

**Fig 4. Three-dimensional plots locations of pieces of metal in pheasant carcasses.** This shows the locations of metal shards (black circles) and whole or near-whole shotgun pellets (red circles) in the carcasses of eight pheasants. The sizes of the plotted symbols are not to scale. The axes are in millimetres. The grid on the base of each diagram shows 50x50 mm squares.

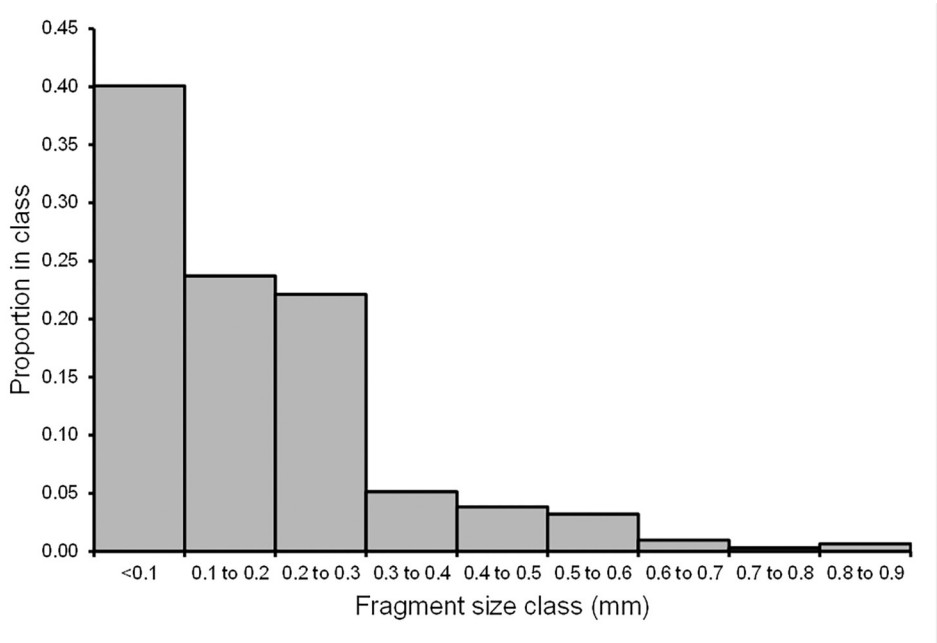

**Fig 5. Size distribution of small metal fragments.** Frequency distribution of the estimated diameters $D$ of small metal fragments <1 mm imaged in micro-CT scans. Bars show the proportions of 312 fragments in each 0.1 mm size class. No fragments in the range 1–2 mm were detected.

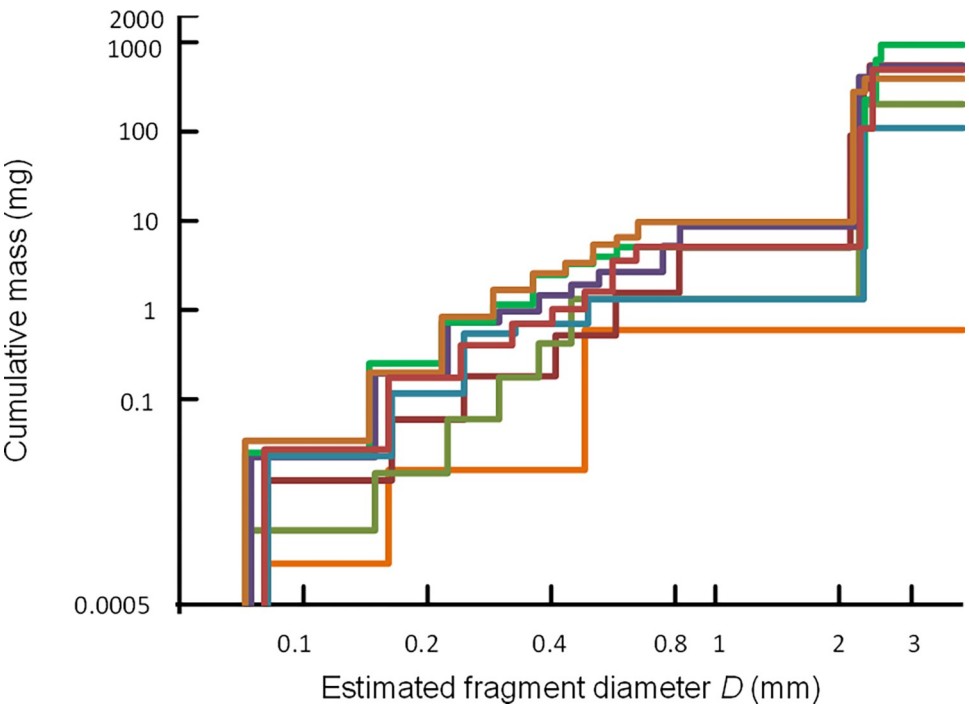

**Fig 6. Cumulative mass of ammunition-derived lead in relation to fragment diameter.** Cumulative mass (mg), estimated from CT voxel counts, expected to be ingested by consumers eating meat from each of eight pheasants in relation to the consumers' potential rejection threshold based upon fragment diameter (mm). We assumed that all fragments smaller than the threshold would be ingested by the consumers and all fragments larger than the threshold would be rejected. The result for each of the pheasants we sampled is represented by a coloured line. Both axes are logarithmic but the values shown are not transformed.

the lead pellets. Hence, our results indicate that there were substantial amounts of metallic lead (maximum value 9.7 mg) present in pheasant carcasses in the form of fragments <1 mm in diameter.

It seems unlikely that consumers of meat from wild-shot pheasant carcasses would detect and reject all of these metal fragments, either during food preparation or when eating. Some lead could be removed during food preparation, such as whole and nearly whole shot. In addition, it has been suggested that metal fragments in meat from near the wound channel could be dissected away and discarded. Kollander et al. [14], reported an experimental study of enhanced meat preparation using carcasses of twenty carrion crows (*Corvus corone*) killed using shotgun ammunition. The breast muscle was removed, and any visibly damaged tissue and shot were removed prior to analysis of the remaining tissue. Two shotgun pellet types were used: a sports shooting shot type sometimes used for small game animals and a standard small game shooting shot. The sports shooting shot resulted in mean tissue lead levels almost 400 times the EU ML (applied to most farm reared meat products) of 100 ppb (wet weight) before cleaning, falling to 14 times the EU ML after cleaning. The game shooting shot type resulted in mean tissue lead levels 1840 times the EU ML before cleaning (as one shot remained in a sample; 1.8 times the EU ML after excluding this sample), dropping to the EU ML following cleaning. Hence, even with very careful butchering of crow breast tissue alone, mean lead levels following cleaning did not consistently drop below the EU ML applied to other meat products. It should also be noted that wild-shot crows are not widely used for food and the results from this study might not be applicable to more frequently consumed bird species.

Pain et al. [17] analysed meals made from gamebird carcasses, including those of pheasants, from which shotgun pellets and large pieces thereof, detected on X-radiographs, were all removed. They found that the arithmetic mean concentration of lead in the meals was 980 ppb (wet weight) per unit weight of meat, which greatly exceeds the maximum level of 100 ppb (wet weight) set for meats from domesticated animals under EU Commission Regulation 1881/2006. Their multiple regression analysis showed that the concentration of lead in the game meat was significantly positively, but non-linearly, related to the number of shotgun pellets in the carcass and also the number of small metal fragments detected on X-radiographs [17]. These results, when combined with our findings from micro-tomography, suggest that the lead they found in meals prepared from carcasses of wild-shot gamebirds from which lead shotgun pellets had been removed was largely derived from the small fragments of metallic lead. We also found these fragments to be numerous and widely distributed in pheasant carcasses. Absorption of metallic lead present as small fragments in the diet has previously been demonstrated experimentally in pigs (*Sus scrofa domestica*) [12] and rats (*Rattus norvegicus*) [13], with the proportion of lead absorbed by rats being substantial for small particles of metallic lead in the size range of those detected by our study of pheasant carcasses [22]. Hence, the correlations observed previously between blood lead levels of humans and their frequency of consumption of meat from birds killed using lead ammunition [11] are likely to have arisen because of absorption of lead in the diet derived from small fragments of projectiles embedded in the meat.

## 5. Conclusion

Most of the mass of ammunition-derived lead identified in the carcasses of wild-shot pheasants we imaged was in the form of whole or nearly-whole shotgun pellets. Although consumers of game meat sometimes ingest shotgun pellets, they probably do so rarely. By contrast, many of the metal fragments we located in pheasant carcasses were too small for it to be likely that consumers would detect and remove them during food preparation or whilst eating. Many fragments were also too distant from the nearest shotgun pellet for it to be practical to detect and remove them without discarding an unrealistically large proportion of otherwise usable meat. Our results indicate that consumers of pheasants and other similar gamebirds are likely to remain exposed to elevated levels of dietary lead in the form of small fragments of metallic lead while lead gunshot continues to be used for hunting, even if more careful food preparation is practiced to remove embedded shotgun pellets and the most damaged tissue.

## Acknowledgments

We thank the University Museum of Zoology, Cambridge UK for assistance with using the Cambridge Biotomography Centre's CT scanner and Matthew Stasiewicz for constructive comments on a previous version.

## Declarations

**Compliance with Ethics Requirements:** This study does not involve any animal experiments. Carcasses of dead birds used in this study were purchased from food retailers and were farmed or legally killed by hunters in the UK.

## Author Contributions

**Conceptualization:** Rhys Green, Mark Taggart, Deborah Pain.

**Data curation:** Rhys Green, Mark Taggart, Keturah Smithson.

**Formal analysis:** Rhys Green.

**Funding acquisition:** Rhys Green.

**Investigation:** Rhys Green, Mark Taggart, Deborah Pain, Keturah Smithson.

**Methodology:** Rhys Green, Mark Taggart, Deborah Pain, Keturah Smithson.

**Project administration:** Rhys Green.

**Resources:** Rhys Green, Mark Taggart.

**Software:** Rhys Green.

**Supervision:** Rhys Green.

**Validation:** Rhys Green.

**Visualization:** Rhys Green.

**Writing – original draft:** Rhys Green.

**Writing – review & editing:** Rhys Green, Mark Taggart, Deborah Pain, Keturah Smithson.

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
