## [Decision Letter · Decision Letter 0]

30 May 2022

PONE-D-22-11623Implications for food safety of the size and location of fragments of lead shotgun pellets embedded in hunted carcasses of small game animals intended for human consumptionPLOS ONE

Dear Dr. Green,

Thank you for submitting your manuscript to PLOS ONE. After careful consideration, we feel that it has merit but does not fully meet PLOS ONE’s publication criteria as it currently stands. Therefore, we invite you to submit a revised version of the manuscript that addresses the points raised during the review process.

PLEASE ADDRESS THE COMMENTS FROM THE REVIEWER.

We look forward to receiving your revised manuscript.

Kind regards,

Juan J Loor

Academic Editor

PLOS ONE

Journal Requirements:

Reviewers' comments:

Reviewer's Responses to Questions

**Comments to the Author**

1. Is the manuscript technically sound, and do the data support the conclusions?

Reviewer #1: Yes

2. Has the statistical analysis been performed appropriately and rigorously? 

Reviewer #1: Yes

3. Have the authors made all data underlying the findings in their manuscript fully available?

Reviewer #1: Yes

4. Is the manuscript presented in an intelligible fashion and written in standard English?

Reviewer #1: Yes

5. Review Comments to the Author

Reviewer #1: Summary

The authors present a study that determines the composition, size distribution, and relative location, of metal shotgun pellets remaining in game pheasant sold for human consumption.

General Comments

Overall, this is a sound study that confirms small shotgun pellets are likely to remain in game birds after reasonable cleaning, that those pellets are likely to be primarily lead, and that those pellets represent enough total lead mass to represent a food safety risk. In particular, one that the authors point out has likely been neglected in internal risk assessments around lead exposure through food. These conclusions are supported by careful, appropriate laboratory analysis to image fragments and then extra and confirm their composition. My comments are primarily to improve the presentation of the manuscript, to make things clearer.

Line Item Comments

65-67: Very good point about the implicit assumption made in Codex around lead exposure.

79 and overall intro: The comment about exposure to lead and absorption caused me, as reader, to question: Would consumers actually eat these small pellets? And even if so, is lead in small shot pellets actually bioavailable/absorbed? The discussion actually address these points well, in 358-363, refs 12, 20-22. I’d suggest moving this up to the intro.

131. I’d suggest adding a header for this section. Something like CT image calibration curves, or control.

161. Is the assumption of approximately spherical fragments fair? Later the authors discuss error in mass estimates based on diameter measures as due to imaging artifacts. But couldn’t an alternative explanation be that fragments are much more like flakes than spheres? Then, if the image detects the longest axis and calls it diameter, that would lead to systematically overestimate mass. (see comment 2 below for more context)

172. Provide a reference to the AES method, if possible. It reads like the authors developed this method from scratch for the paper.

208-210. In addition to the r value in the text and legend, it would be helpful to report the slope and intercept of the fitted regression. The slope would be a summary statistic for the point about correlation of mean diameters. Then, blooming comes up in 211.

Figure 1. Please add the regression line equations to the figure. At min, the equation for the regression where the intercept is constrained to the origin (a line with no intercept). But I’d also suggest adding the y=mx+b line relevant to 208-210.

246: ‘Whole or near-whole’ seems like the wrong term for fragments > 2 mm diameter. If the mean is 3.5 fragments, it doesn’t seem logical they could all be near-whole. Maybe just use ‘large’ and ‘small’ fragments?

246-248. Not clear the means are the right summary statistics for the data, given that the distributions are not clear. It might help to make boxplots for the fragment sizes in all the birds, maybe stratified by small and large, maybe not. Seems like individual carcass variation is great.

Figure 5. Put in the legend that no fragments were detected 1-2 mm. Reading the paper I wondered why the x-axis stopped at 1 mm not 2 mm.

Figure 6. This cumulative exposure graph is great. It definitely shows the point there is a risk represented by small fragments, < 2 mm, that can cause birds to exceed guidance levels. But, it also shows a second point not addressed in the text. It is very important that all large fragments are identified and removed. Seems like even one large, > 2 mm, fragment would cause lead exposure in the order of grams, based on the log scale axis. And it’s not obvious to me all large fragments would be identified by cleaning (obviously, given your data) or eating (people eat fast, get distracted, mis-attribute to gristle/bone, kids might not realize). I’d suggest adding a small discussion point around this observation, if the authors feel it has merit.

6. PLOS authors have the option to publish the peer review history of their article (what does this mean?). If published, this will include your full peer review and any attached files.

Reviewer #1: **Yes: **Matthew Stasiewicz

---

## [Author Response · Author response to Decision Letter 0]

5 Jul 2022

PONE-D-22-11623

Implications for food safety of the size and location of fragments of lead shotgun pellets embedded in hunted carcasses of small game animals intended for human consumption

RESPONSE TO REVIEWERS

After resubmitting the revised version of this MS, I received a message on 29 June 2022 saying that our previous resubmitted MS PONE-D-22-11623R1 was incomplete. The missing item was said to be the manuscript with Track Changes shown. I checked the compiled pdf PONE-D-22-11623R1 at Editorial Manager and found that manuscript with Track Changes was already there on pages 42-66 of that pdf. I explained this in the Comments box and in emails to PLOS ONE, but to no effect. Editorila Manager still says that this MS in ‘incomplete submission being revised’. So that is why I am now resubmitting the same files as before over again. I hope that it works OK this time.

AUTHORS’ OVERALL RESPONSE

We thank the editor for the opportunity to revise our submission and the reviewer for constructive comments. We have labelled each substantive comment (e.g. COMMENT 1.1, COMMENT 1.2, etc) and then followed each with a response (labelled ‘RESPONSE’).

Reviewers' comments:

Reviewer's Responses to Questions

Comments to the Author

1. Is the manuscript technically sound, and do the data support the conclusions?

Reviewer #1: Yes

2. Has the statistical analysis been performed appropriately and rigorously?

Reviewer #1: Yes

3. Have the authors made all data underlying the findings in their manuscript fully available?

Reviewer #1: Yes

4. Is the manuscript presented in an intelligible fashion and written in standard English?

Reviewer #1: Yes

5. Review Comments to the Author

COMMENT 1.1

Reviewer #1: Summary

The authors present a study that determines the composition, size distribution, and relative location, of metal shotgun pellets remaining in game pheasant sold for human consumption.

General Comments

Overall, this is a sound study that confirms small shotgun pellets are likely to remain in game birds after reasonable cleaning, that those pellets are likely to be primarily lead, and that those pellets represent enough total lead mass to represent a food safety risk. In particular, one that the authors point out has likely been neglected in internal risk assessments around lead exposure through food. These conclusions are supported by careful, appropriate laboratory analysis to image fragments and then extra and confirm their composition. My comments are primarily to improve the presentation of the manuscript, to make things clearer.

RESPONSE:

We thank the reviewer for this positive assessment and welcome the recommendations for improvements.

Line Item Comments

COMMENT 1.2

65-67: Very good point about the implicit assumption made in Codex around lead exposure.

RESPONSE:

We appreciate the reviewer’s comment on this. As we noted on line 65, there appears to be no formal written argument accompanying the omission from the Codex Alimentarius of ingestion of lead ammunition fragments as a pathway for dietary exposure to lead. It is helpful to see that the reviewer accepts our point about this and finds it useful. We have therefore not modified the text in this regard.

COMMENT 1.3

79 and overall intro: The comment about exposure to lead and absorption caused me, as reader, to question: Would consumers actually eat these small pellets? And even if so, is lead in small shot pellets actually bioavailable/absorbed? The discussion actually address these points well, in 358-363, refs 12, 20-22. I’d suggest moving this up to the intro.

RESPONSE:

This is a helpful suggestion. As proposed by the reviewer, we have added text to the Introduction to summarise evidence that lead derived from ammunition is both ingested and absorbed. The added text is as follows:

‘However, observational studies of humans found a positive correlation between the concentration of lead in blood plasma and the frequency of consumption of the meat of wild animals killed using lead ammunition [11]. This suggests that some lead derived from ammunition is both ingested and absorbed by human consumers. Absorption of lead from ingested particles of the metal has also been demonstrated experimentally in non-human mammals [12, 13]. If the apparent assessment of a negligible potential hazard from ammunition-derived lead in game meat by the Codex Alimentarius and the European Union’s Regulation (EC) No 1881/2006 is incorrect, this may have negative consequences for public health. Removing small and widely-distributed fragments of ammunition-derived metallic lead from meat is difficult [14], so dietary exposure to ammunition-derived lead and its absorption and significance for public health may be greater than was previously assumed.’

We have left the previous text about absorption of metallic lead by humans and non-human mammals in the Discussion as it stood, rather than moving it to the Introduction as the reviewer suggested. That is because the Discussion text makes a more detailed point about the relationship between absorption and particle size.

COMMENT 1.4

131. I’d suggest adding a header for this section. Something like CT image calibration curves, or control.

RESPONSE:

This is a useful suggestion. We have added three subheadings to divide up section 2.2: ‘2.2.1 Wild-shot pheasants, ‘2.2.2 Calibration using domestic fowl ‘, ‘2.2.3 Estimation of fragment size‘.

COMMENT 1.5

161. Is the assumption of approximately spherical fragments fair? Later the authors discuss error in mass estimates based on diameter measures as due to imaging artifacts. But couldn’t an alternative explanation be that fragments are much more like flakes than spheres? Then, if the image detects the longest axis and calls it diameter, that would lead to systematically overestimate mass. (see comment 2 below for more context).

RESPONSE:

We think that the reviewer’s comment is not entirely correct, though we agree that it is useful to clarify the point raised. The reviewer has taken it that we measured the longest axis of an imaged fragment to determine its apparent diameter, but that is not what we did. As we stated previously on line 160, we counted the number of high-intensity voxels in an imaged fragment to determine volume V and then estimated apparent diameter as AD = 2∛(3V/4π). So our method obtains an average diameter for a non-spherical object and not the length of its longest axis. We accept though that the assumption introduces some error, though not the bias (overestimation) that the reviewer proposed. We have therefore added the following text to explain that: ‘This procedure would introduce some inaccuracy into the estimation of the mean diameter of non-spherical fragments, but the dimensions of clumps of high-intensity voxels did not suggest any marked asymmetry, so we think that the effect of this is likely to be minor.’

COMMENT 1.6

172. Provide a reference to the AES method, if possible. It reads like the authors developed this method from scratch for the paper.

RESPONSE:

We have added an appropriate reference to further details of the ICP-AES method, which was described in ref. 4 of the original listing, but not cited as such.

COMMENT 1.7

208-210. In addition to the r value in the text and legend, it would be helpful to report the slope and intercept of the fitted regression. The slope would be a summary statistic for the point about correlation of mean diameters. Then, blooming comes up in 211.

Figure 1. Please add the regression line equations to the figure. At min, the equation for the regression where the intercept is constrained to the origin (a line with no intercept). But I’d also suggest adding the y=mx+b line relevant to 208-210.

RESPONSE:

This comment is helpful in alerting us to the fact that we unintentionally glossed over our reasons for using a regression model in which apparent diameter was assumed to be directly proportional to true diameter and therefore for using a regression constrained to pass through the origin, rather than an ordinary regression with non-zero intercept. We have responded by adding text describing a formal comparison of these two alternative regression models. This comparison supports our a priori expectation that apparent diameter is in direct proportion to true diameter. Although we now give the regression parameters for the model with a non-zero intercept in the main text (as requested by the reviewer), we do not show this model in Figure 1, because the model with zero intercept is better supported by the data. The new text added to describe these results is as follows:

‘We expected a priori that AD and D would be directly proportion to each other and therefore that a least-squares regression constrained to pass through the origin (D = cAD) would give a good description of the relationship for prediction of D from observed values of AD determined from images of fragments in pheasant carcasses. This expectation was supported by a comparison of the residual sums of squares from this model with that from the ordinary least-squares regression model in which both the intercept a and the slope b were estimated (D = a + bAD), with a = -0.0981 and b = 0.854. Comparison of the two models indicated that inclusion of the intercept term a did not result in a significant improvement of model fit (F1,8 = 3.66, P = 0.092). The estimate of c for the regression model constrained to pass through the origin was 0.724 (standard error = 0.031).’

COMMENT 1.8

246: ‘Whole or near-whole’ seems like the wrong term for fragments > 2 mm diameter. If the mean is 3.5 fragments, it doesn’t seem logical they could all be near-whole. Maybe just use ‘large’ and ‘small’ fragments?

RESPONSE:

We think that there has been a misunderstanding here. All theCT-imaged radio-dense objects >2mm were near-spherical and those recovered and analysed by ICP-AES were all found to be composed principally of lead (see section 3.1). We therefore conclude that the imaged objects >2mm were whole or nearly whole lead shotgun pellets. We have redrafted this section to clarify this as follows:

‘It is likely that all of the large, apparently near-spherical, radio-dense >2 mm diameter objects we observed on the CT images were whole or nearly-whole shotgun pellets, which we identified by ICP-AES analysis as being composed principally of lead (see section 3.1). The most frequently used sizes of shotgun pellets used to kill pheasants in the UK are #4, #5, and #6 (diameters 2.6 – 3.1 mm). The CT images identified a mean of 3.5 shotgun pellets per pheasant carcass (SE = 0.8; range 0 to 7) and a mean of 39.0 smaller (<2 mm diameter) metal fragments per carcass (SE = 10.7; range 3 to 68).’

COMMENT 1.9

246-248. Not clear the means are the right summary statistics for the data, given that the distributions are not clear. It might help to make boxplots for the fragment sizes in all the birds, maybe stratified by small and large, maybe not. Seems like individual carcass variation is great.

RESPONSE:

We have moved the information on the ranges of fragment and pellets counts per carcasses so that it is given alongside the means and SEs. We think that this makes the level of variation among carcasses quite clear. Both Figures 5 and 6 give information on the size distribution of fragments and Fig 6 does so separately for each carcass. Hence, we think that the extra boxplots suggested by the reviewer are not needed.

COMMENT 1.10

Figure 5. Put in the legend that no fragments were detected 1-2 mm. Reading the paper I wondered why the x-axis stopped at 1 mm not 2 mm.

RESPONSE:

Done, though we did already say this at lines 269-270 of the main text.

COMMENT 1.11

Figure 6. This cumulative exposure graph is great. It definitely shows the point there is a risk represented by small fragments, < 2 mm, that can cause birds to exceed guidance levels. But, it also shows a second point not addressed in the text. It is very important that all large fragments are identified and removed. Seems like even one large, > 2 mm, fragment would cause lead exposure in the order of grams, based on the log scale axis. And it’s not obvious to me all large fragments would be identified by cleaning (obviously, given your data) or eating (people eat fast, get distracted, mis-attribute to gristle/bone, kids might not realize). I’d suggest adding a small discussion point around this observation, if the authors feel it has merit.

RESPONSE:

We agree that this is a useful point and have added some new text at the beginning of the Conclusions section to make it. As follows:

‘Most of the mass of ammunition-derived lead identified in the carcasses of wild-shot pheasants we imaged was in the form of whole or nearly-whole shotgun pellets. Although consumers of game meat sometimes ingest shotgun pellets, they probably do so rarely. By contrast, many of the metal fragments we located in pheasant carcasses were too small for it to be likely that consumers would detect and remove them during food preparation or whilst eating.’

---

## [Decision Letter · Decision Letter 1]

28 Jul 2022

Implications for food safety of the size and location of fragments of lead shotgun pellets embedded in hunted carcasses of small game animals intended for human consumption

PONE-D-22-11623R1

Dear Dr. Green,

We’re pleased to inform you that your manuscript has been judged scientifically suitable for publication and will be formally accepted for publication once it meets all outstanding technical requirements.

Kind regards,

Juan J Loor

Academic Editor

PLOS ONE

Additional Editor Comments (optional):

Reviewers' comments:

Reviewer's Responses to Questions

**Comments to the Author**

1. If the authors have adequately addressed your comments raised in a previous round of review and you feel that this manuscript is now acceptable for publication, you may indicate that here to bypass the “Comments to the Author” section, enter your conflict of interest statement in the “Confidential to Editor” section, and submit your "Accept" recommendation.

Reviewer #1: All comments have been addressed

2. Is the manuscript technically sound, and do the data support the conclusions?

Reviewer #1: (No Response)

3. Has the statistical analysis been performed appropriately and rigorously? 

Reviewer #1: (No Response)

4. Have the authors made all data underlying the findings in their manuscript fully available?

Reviewer #1: (No Response)

5. Is the manuscript presented in an intelligible fashion and written in standard English?

Reviewer #1: (No Response)

6. Review Comments to the Author

Reviewer #1: No remaining issues. Though odd to be acknowledged in the paper for peer review. No objections to that though.

7. PLOS authors have the option to publish the peer review history of their article (what does this mean?). If published, this will include your full peer review and any attached files.

Reviewer #1: **Yes: **Matthew Stasiewicz

---

## [Editor Report · Acceptance letter]

2 Aug 2022

PONE-D-22-11623R1 

Implications for food safety of the size and location of fragments of lead shotgun pellets embedded in hunted carcasses of small game animals intended for human consumption 

Dear Dr. Green:

I'm pleased to inform you that your manuscript has been deemed suitable for publication in PLOS ONE. Congratulations! Your manuscript is now with our production department. 

Kind regards, 

on behalf of

Dr. Juan J Loor 

Academic Editor

PLOS ONE